# Inflammatory Markers after Switching to a Dual Drug Regimen in HIV-Infected Subjects: A Two-Year Follow-Up

**DOI:** 10.3390/v14050927

**Published:** 2022-04-28

**Authors:** Matteo Vassallo, Jacques Durant, Roxane Fabre, Laurene Lotte, Audrey Sindt, Annick Puchois, Anne De Monte, Renaud Cezar, Pierre Corbeau, Christian Pradier

**Affiliations:** 1Department of Internal Medicine/Infectious Diseases, Cannes General Hospital, 06400 Cannes, France; 2Unité de Recherche Clinique Cote d’Azur (UR2CA), URRIS, Centre Hospitalier Universitaire Pasteur 2, 06300 Nice, France; 3Department of Infectious Diseases, University Côte d’Azur, 06108 Nice, France; durant.j@chu-nice.fr; 4Department of Public Health, L’Archet Hospital, University of Nice, 06107 Nice, France; fabre.r@chu-nice.fr (R.F.); pradier.c@chu-nice.fr (C.P.); 5CoBteK (Cognition-Behaviour-Technology) Lab, FRIS-University Côte d’Azur, 06108 Nice, France; 6Multipurpose Laboratory, Cannes General Hospital, 06400 Cannes, France; la.lotte@ch-cannes.fr (L.L.); a.sindt@ch-cannes.fr (A.S.); a.hugot@ch-cannes.fr (A.P.); 7Laboratory of Virology, Nice University Hospital, University Côte d’Azur, 06108 Nice, France; demonte.a@chu-nice.fr; 8Laboratory of Immunology, Nimes University Hospital, 30029 Nimes, France; renaud.cezar@chu-nimes.fr (R.C.); pierre.corbeau@chu-nimes.fr (P.C.)

**Keywords:** HIV, successful treatment, simplification strategies, inflammation, macrophage activation

## Abstract

Objective: Immunadapt is a study evaluating the impact of combination antiretroviral treatment (cART) simplification on immune activation. We previously showed that switching to dual therapies could be associated six months later with macrophage activation. Followup continued up to 24 months after treatment simplification. Materials and Methods: Immunadapt is a prospective single arm study of successfully treated subjects simplifying cART from triple to dual regimens. Before cART change, at 6 months, and between 18 and 24 months following the switch, we measured IP-10, MCP-1, soluble CD14 (sCD14), soluble CD163 (sCD163), and lipopolysaccharide binding protein. Patients were stratified according to lower or greater likelihood of immune activation (CD4 nadir < 200, previous AIDS-defining event or very-low-level viremia during follow-up). Variables were compared using matched Wilcoxon tests. Results: From April 2019 to September 2021, 14 subjects were included (mean age 60 years, 12 men, 26 years since HIV infection, CD4 nadir 302 cells/mm^3^, 18 years on cART, 53 months on last cART). Twenty-one months following the switch, all but one subject maintained their viral load < 50 cp/mL. One subject had two viral blips. For the entire population, the sCD163 values increased significantly from baseline (+36%, *p* = 0.003) and from 6 months after the switch. The other markers did not change. After 6 months, the sCD163 increase was more pronounced in subjects with greater likelihood of immune activation (+53% vs. +19%, *p* = 0.026) Conclusions: cART simplification to dual therapy was associated with macrophage activation despite successful virological control after almost two years’ follow-up. This was more pronounced in those at risk of immune activation.

## 1. Introduction

The life expectancy of people living with HIV (PLH) has dramatically improved since the introduction of triple-drug antiretroviral regimens [1].

However, the long-term potential side effects of combination antiretroviral therapy (cART) have led to the introduction of guidelines recommending dual therapies, generally consisting in the association of an integrase inhibitor with another molecule, such as a nucleoside- or a non-nucleoside transcriptase inhibitor. Dual therapies have proven their non-inferiority in terms of virological control when compared to triple c-ART [2,3,4,5].

However, data on the consequences on the inflammatory biomarkers of switching to dual therapies are contradictory, and few studies have focused on the effect on the immune activation of such a switch. The TANGO and SWORD trials, together with a recent systematic review, found no consistent pattern of change in inflammatory biomarkers with two-drug regimens [6,7,8]. However, Serrano-Villar et al. showed that levels of three plasma inflammatory markers were increased in patients receiving dual therapy [9]. Moreover, we recently showed that switching to dual therapies could be associated 6 months later with a significant increase in sCD163, a well-known marker of macrophage activation, in subjects with a low CD4 nadir, previous AIDS, or residual viremia during follow-up [10]. Among the hypotheses explaining such macrophage activation, it has been suggested that reduced drug pressure in the reservoir may potentially expose the virus to sub-optimal antiretroviral concentrations [11]. 

Immunadapt is a prospective study aiming to assess the impact on immune activation and inflammatory markers of switching from a triple-drug to a dual-drug therapy. Followup was continued up to 2 years, and the results are shown here. 

## 2. Materials and Methods

### 2.1. Study Design

Immunadapt is a single-arm prospective study initiated in April 2019. Its aim is to measure the impact on immune activation markers of switching from a triple-drug to a dual-drug regimen. Patients were selected among those routinely followed in the Department of Internal Medicine in Cannes General Hospital and the Department of Infectious Diseases in Nice University Hospital.

Each patient meeting the inclusion criteria was offered to participate in this study.

Recruitment continued until achieving 20 subjects, which was the sample size estimated necessary for the analysis. 

HIV-1 infected subjects on stable and successful cART (viral load < 50 copies/mL for at least 6 months, measured with Xpert^©^ viral load or Aptima HIV Quant Dx, Hologic, Tremblay-en-France, France), switching from a triple-drug to a dual-drug regimen as a simplification strategy, were included.

Subjects who were not on stable and successful cART, or those whose treatment included a different number of compounds, such as those switching from a quadruple- to a triple-drug regimen could not participate.

This study was approved by the Paris Ethics Committee (Comité de Protection des Personnes, Ile de France IV), and patients gave written informed consent to participate.

The study was initially designed for a follow-up period of 6 months. However, based on our findings at that point, [10], it was decided to continue follow-up until 24 months. An extension of the study was therefore requested and approved by the Ethics Committee. The included individuals gave their consent to such an extension.

### 2.2. Markers of Immune Activation

Measurement of immune activation was performed just before the cART switch, approximately 6 months later, and between 18 and 24 months later. IP-10 and MCP-1 were measured using ProcartaPlex ImmunoAssays (Life Technologies SAS, Waltham, MA USA), while sCD163 and sCD14 (Quantikine ELISA kit, Biotechne, Minneapolis, MI, USA) and lipopolysaccharide binding protein (LBP, Enzo Life Sciences, Villeurbanne, France) were measured with ELISA kits. As the study was initially designed for a 6-month follow-up, samples collected at inclusion and 6 months after the switch were frozen at −20° and were all analyzed simultaneously. An extension of the study was subsequently requested, and samples collected between 18 and 24 months were analyzed at a later date.

### 2.3. Demographic Parameters and Background Measurements

Demographic and main viro-immunological characteristics were collected. Prior antiretroviral therapy was also recorded. Regimens were classified according to the following categories: nucleoside reverse transcriptase inhibitors (NRTI), non-nucleoside reverse transcriptase inhibitors (NNRTI), integrase strand transfer inhibitor (INSTI), and protease inhibitors (PI).

During follow-up, plasma viral load was quantified 2 months after the switch. Viro-immunological parameters were measured, together with immune activation markers, approximately 6 months after treatment simplification and at the end of follow-up, i.e., between 18 and 24 months from the switch. 

Viral blips were defined as transient low-level viremia (LLV), between 50 and 500 copies/mL, preceded and followed by suppression, i.e., below the quantification limit of the assay [12]. To assess the impact on immune activation, we also checked for very-low-level viremia (VLLV) detected at inclusion or during follow-up. VLLV was defined as viremia < 50 copies/mL detected by clinical assays with quantification cutoffs of <50 copies/mL [13]. LLV and VLLV during the two years prior to inclusion were also checked according to patients’ files, in order to evaluate the impact of previous residual viremia on immune activation.

During follow-up, clinical events and newly prescribed co-medications were also recorded. 

### 2.4. Statistical Analysis

The main demographic and viro-immunological characteristics of the population are described. 

Baseline was defined as the day of the switch from a triple to a dual cART regimen. Differences in parameter values were measured from baseline to 6 months from the switch, from 6 months to 18–24 months, and from baseline to 18–24 months.

We first measured changes from baseline to 6 months and 18–24 months after treatment simplification for the entire study population. Considering the small sample size and the fact that CD4/CD8 ratio at inclusion and IP-10, MCP-1, and LBP trajectories did not follow a normal distribution (Shapiro-Wilk test of normality), variables were compared using the matched Wilcoxon’s test.

Patients were then stratified into those with lower or higher likelihood of immune activation, the latter defined by at least one of the following parameters: low CD4 nadir (i.e., <200 cc/mm^3^), prior AIDS-defining condition, or VLLV during follow-up [14]. Differences were measured with the Wilcoxon-Mann-Whitney test. Statistical analysis was performed using R 4.0.3 software.

## 3. Results

From April 2019 to September 2021, 20 subjects participated in this study for the initial 6-month follow-up period (90% men, mean age 57 years, 25 years since known HIV infection, CD4 cell count at inclusion 666 cells/mm^3^, CD4/CD8 ratio 0.94, CD4 nadir 326 cells/mm^3^, 18 years on cART, 15% with prior history of AIDS, 6 cART regimens received) and 14 of them continued to participate for a mean period of 21 months (Table 1). Main triple c-ART before the switch included 2 NRTI and 1 INSTI or 2 NRTI and 1 NNRTI (Table 1).

The main reasons for switching antiretroviral therapy were either to reduce the number of pills or to limit the potential long-term side effects of antiretrovirals. Prescribed dual cART consisted in a combination of Dolutegravir and Rilpivirine for 11 patients and Dolutegravir and Lamivudine for the remaining 3 patients.

No significant co-medications with potential interference on immune activation were prescribed during follow-up. One individual suffered an ischemic stroke and one a pulmonary embolism, approximately six months after the switch. No clinical event nor intolerance to treatment occurred among the remaining 12 individuals.

Among the 14 patients completing the follow-up, all but one maintained a viral load below 50 cp/mL. However, the number of those with VLLV increased from two individuals at baseline to four at 6 months and to six at the end of follow-up.

One subject had two viral blips during follow-up (66 cp/mL and 52 cp/mL), requiring a switch of therapy to a triple-drug regimen. The six individuals not completing the extended follow-up period maintained their viral load below 50 cp/mL all along the 21 month-period after the switch (data not shown).

The characteristics of patients and trends in immune activation markers are described in Table 2. For the entire population, the sCD163 values increased significantly from baseline (+36%, *p* = 0.003) and from 6 months after the switch (Figure 1). Among the other immune activation markers, a trend was observed for an improvement in MCP-1 values after the switch. Of note, this improvement tended to be lower in subjects with higher likelihood of immune activation (Table 2). The other immune activation markers did not change.

As observed at the initial 6-month endpoint, differences in the sCD163 trajectory were greater among subjects with greater likelihood of immune activation (+53% vs. +19%, *p*= 0.026, Table 2, Figure 2). The results did not change even excluding the subject with two viral blips (data not shown). Patients with just VLLV during the follow-up as a risk factor for immune activation did not have a significantly higher increase in sCD163 (data not shown).

No differences in the sCD163 trajectory were found between individuals switching to Dolutegravir and Rilpivirine and those switching to Dolutegravir and Lamivudine (data not shown).

## 4. Discussion

In a population of successfully treated subjects switching from a triple-drug to a dual drug-HIV regimen as a simplification strategy, we found a significant increase in macrophage activation. Such immune activation continued to increase after a follow-up of almost 2 years and was higher in patients with a low CD4 nadir, a previous AIDS-defining event, or detectable residual viremia during follow-up. Macrophage activation has been linked to several non-AIDS-related complications, such as neurological, cardiovascular, and hepatic disorders [15,16,17,18].

In particular, sCD163 is considered one of the best markers for atherosclerotic plaque formation in HIV-infected patients as well as among the general population [17]. Moreover, macrophage activation plays a central role in HIV inflammatory disease. Indeed, different initiators of inflammation, such as microbial translocation, loss of regulatory responses and HIV replication, are responsible for chronic activation of innate immunity, contributing to clinical aging [19].

Interestingly, the ANRS-ETRAL trial already showed an increase in sCD163 when switching from a boosted PI-containing regimen to Raltegravir and Etravirine dual therapy [20]. In addition, Llibre et al. found an increase in sCD163 together with Il-6 during the late-phase switch to Dolutegravir and Rilpivirine, although no consistent pattern of change across inflammatory biomarkers was observed [7].

The reasons for such macrophage activation in our study require clarification. As LBP values did not correlate with the trajectory of sCD163, we do not think that microbial translocation could be associated with such macrophage activation. We suggest at least two hypotheses: the first one is that the relatively low penetration of integrase inhibitors in lymph nodes, associated with the suspension of one NRTI, could be associated with insufficient drug pressure in reservoirs, thus facilitating the resurgence of HIV replication [11]. The second hypothesis is that the discontinuation of a two NRTI-containing drug-regimen could be associated with a reduced capacity to inhibit cell-to-cell transmission of the virus. Indeed, HIV infection of target cells with cell-free viral particles is just one pathway for viral spread in the host; this can also happen through the so-called virological synapse. In this case an infected donor cell establishes direct contact with a target cell, enabling the transfer of viral material [21]. While PI effectively block cell-to-cell spread of HIV between T-cells [22] and little is known of second-generation INSTI in this respect, blocking of cell-to-cell transmission is significantly less effective without the combination of two NRTI compounds [23]. The fact that the number of patients with residual viremia increased during follow-up could be interpreted as a possible consequence of these two hypotheses. However, very prudent conclusions should be drawn, as the significance of VLLV is still unclear, and, to our knowledge, there are no other reports of a possible increase in residual viremia after switching to a dual-drug regimen [13].

As has been reported from previous studies, including ours [10,24], we found different trajectories for sCD14 and sCD163, thus confirming that these two markers follow distinct pathways of macrophage activation, the latter being more specifically associated with residual viremia through its shedding from quiescent CD4+ T cells [25]. Moreover, a trend was observed for an improvement in MCP-1 values after the switch, and other studies already showed different trajectories compared with sCD163 [26].

The limitations of this study include the small number of subjects and its single arm design. A comparative arm with patients continuing a triple-drug regimen would shed more light on the effect of treatment simplification on immune activation. However, as no other medical interventions were recorded during the follow-up, in our view, it is unlikely that the trajectory of sCD163 could be explained by other factors than treatment modification. With the absence of a comparative arm evaluating the trajectory of sCD163 in subjects not changing therapy, we could say that its increase is associated with physiological aging. However, in our view, this hypothesis is very unlikely, considering the important increase we found over a short period, and also that Kroeze et al. did not find any change in sCD163 in patients on stable treatment [27]. In addition, Knudsen et al. showed that each quartile increase in sCD163 was associated with a 35% increased risk of death over a 10-year period of follow-up [28]. Moreover, although the analysis was performed by a single laboratory, strictly following the same procedures, the biomarkers were measured at two different time points, firstly the baseline-to 6-month samples and then those collected between 18 and 24 months. Unfortunately, we did not measure the HIV-DNA viral load, which would add important information about the viral replication in the cellular reservoir and would have been interesting to correlate with the sCD163 trajectory. In addition, it would been interesting to correlate macrophage activation with the number of cycling monocytes expressing Ki67.

In conclusion, simplification to dual-drug HIV therapy was associated with macrophage activation despite successful virological control after almost 2 years’ follow-up. Such an increase was higher in subjects with greater likelihood of immune activation. Further studies should shed more light on the long-term clinical impact of such macrophage activation.

## Figures and Tables

**Figure 1 viruses-14-00927-f001:**
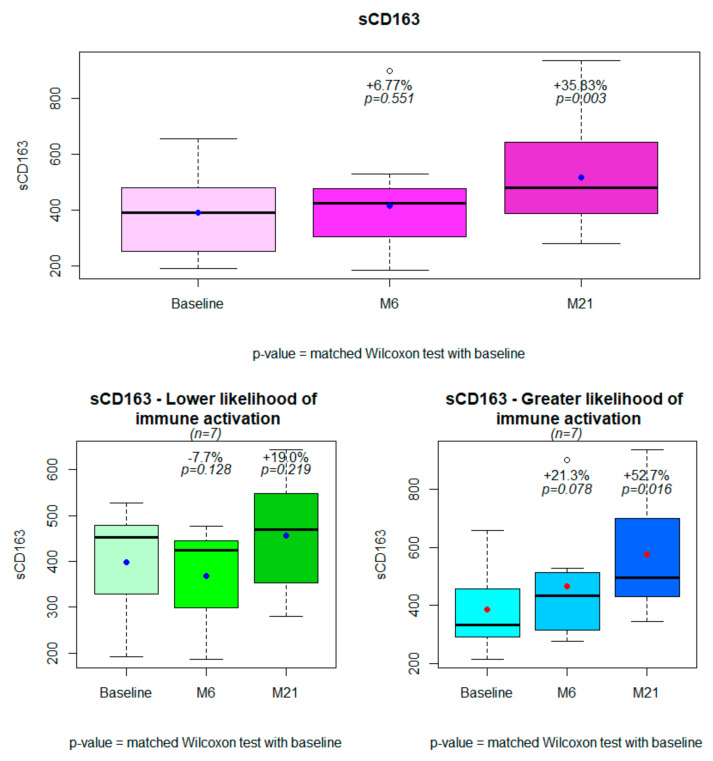
Changes in sCD163 from baseline to 6 months and 21 months after treatment simplification for the entire population (*n* = 14) and in subjects with lower (*n* = 7) or greater (*n* = 7) likelihood of immune activation.

**Figure 2 viruses-14-00927-f002:**
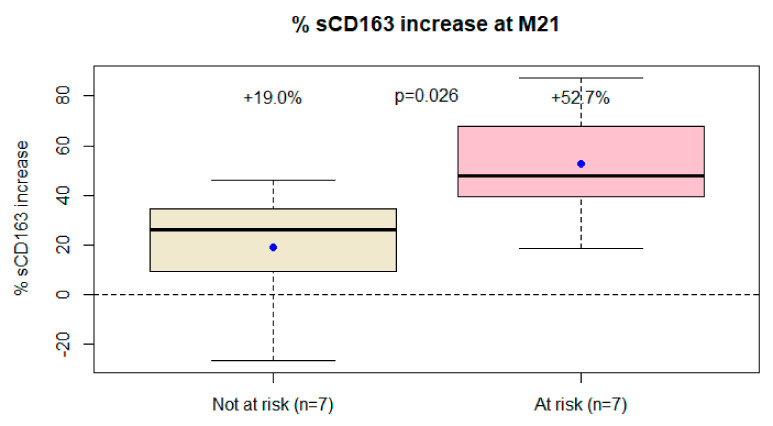
Differences in sCD163 trajectory from baseline to the end of follow-up between patients at risk of immune activation (i.e., CD4 nadir < 200 cc/mm^3^, previous AIDS, or very low-level viremia during the follow-up) and those not at risk.

**Table 1 viruses-14-00927-t001:** Characteristics of patients included.

	N (%) or Median [Q1–Q3]
Number of patients	14
Male gender	12 (86%)
Age (years)	56.7 [54.6; 63.9]
Years since HIV infection	27.0 [20.7; 30.2]
Comorbid conditions	
Hypertension	8 (57%)
Dyslipidemia	4 (29%)
Hepatitis C coinfection	2 (14%)
CD4 cell count at inclusion (cc/mm^3^)	645.5 [496.8; 713.8]
CD8 cell count at inclusion (cc/mm^3^)	687.0 [612.0; 893.0]
CD4/CD8 ratio at inclusion	0.77 [0.67; 1.10]
CD4 nadir (cc/mm^3^)	287.0 [235.3; 352.5]
Years on cART	18.5 [15.0; 22.0]
cART regimens received	7.0 [3.0; 7.8]
Months on current cART	54.0 [37.5; 65.0]
cART received when switching 2 NRTI + 1 INSTI 2 NRTI + 1 NNRTI Other	8 5 1
Dual cART prescribed at inclusion: DTG + RPV DTG + 3TC	11 3

**Table 2 viruses-14-00927-t002:** Patient characteristics and differences according to likelihood of immune activation.

	All Patients *n* = 14		Likelihood of Immune Activation				
			Lower-*n* = 7		Greater-*n* = 7		
	Median	[Q1; Q3]	Median	[Q1; Q3]	Median	[Q1; Q3]	*p*-value *
Age (years)	56.7	[54.6; 63.9]	55.5	[54.3; 58.3]	62.4	[55.9; 70.1]	0.259
Years since HIV infection	27.0	[20.7; 30.2]	27.4	[19.1; 29.2]	26.7	[23.2; 31.2]	0.654
CD4/CD8 ratio inclusion	0.8	[0.7; 1.1]	0.7	[0.6; 0.9]	0.8	[0.8; 1.1]	0.383
Years on cART	18.5	[15.0; 22.0]	16.0	[14.5; 22.0]	20.0	[16.0; 22.5]	0.653
Number of regimens received	7.0	[3.0; 7.8]	6.0	[2.0; 7.0]	7.0	[6.5; 8.0]	0.295
**Median changes from inclusion to 6 months after the switch**							
CD4/CD8 (%)	−1.5	[−7.1; 10.2]	5.9	[−1.6; 9.6]	−5.9	[−10.7; 8.1]	0.628
sCD14 (%)	−15.0	[−24.8; −9.9]	−17.7	[−34.7; −8.4]	−12.3	[−22.0; −11.3]	0.295
sCD163 (%)	0.7	[−9.6; 29.6]	−10.0	[−12.2; 0.7]	29.9	[10.8; 33.7]	0.017
LBP (%)	8.5	[−20.0; 25.9]	−16.8	[−19.1; 17.7]	15.9	[−13.0; 81.6]	0.456
MCP-1 (%)	−19.9	[−24.3; −3.3]	−20.6	[−23.2; −11]	−8.4	[−24.7; 6.1]	0.902
IP-10 (%)	−29.5	[38.2; −19.0]	−32.1	[−37; −24.8]	−25.7	[−40.9; −9.0]	0.383
**Median changes from inclusion to 21 months after the switch**							
CD4/CD8 (%)	−1.8	[−11.3; 7.9]	7.0	[−3.8; 10.5]	−10.8	[−14.3; 1.5]	0.259
sCD14 (%)	−63.7	[−66.0; −58.2]	−57.7	[−63.9; −52.2]	−65.1	[−66.5; −63.4]	0.128
sCD163 (%)	39.5	[20.3; 47.5]	26.0	[9.4; 34.3]	47.9	[39.6; 67.8]	0.026
LBP (%)	33.0	[−2.7; 57.8]	8.4	[−10.8; 40.5]	61.6	[15.9; 69.3]	0.128
MCP-1 (%)	−61.4	[−64.9; −20.6]	−63.7	[−67.9; −61.8]	−41.2	[−60.1; 8.3]	0.097
IP-10 (%)	−57.5	[−68.7; −43.6]	−63.4	[−75.7; −53.0]	−53.3	[−57.5; −25.3]	0.209

* Wilcoxon-Mann-Whitney test. Greater likelihood of immune activation: at least one of the following parameters: low CD4 nadir, previous AIDS stage, or very low-level viremia during follow-up. cART: combination antiretroviral treatment. LBP: Lipopolysaccharide Binding Protein. MCP 1: Monocyte Chemo-attractant Protein-1. IP 10: Interferon γ protein 10.

## Data Availability

The original contributions presented in the study are included in the article; further inquiries can be directed to the corresponding authors.

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
