# Peer review of "Inflammatory Markers after Switching to a Dual Drug Regimen in HIV-Infected Subjects: A Two-Year Follow-Up"

_viruses, 2022, doi:10.3390/v14050927_

Round 1

Reviewer 1 Report

In this paper the authors are following inflammation and activation blood markers in patients who underwent cART modification from traditional Tri-therapy to dual therapy. The authors are reporting soluable inflammation markers quantification before cART switch, then at 6 month later and 18 to 24 months later. I would like to offer a few suggestions and have a few questions to hopefully improve this manuscript.

1- We started with 20 patients, but 6 were excluded because of non following up to 21 months. There is no details on how many time point the authors were able to get form those 6 patients but it would have been great to add them to the study if there is any data at the 6 months time point.

2- Did the patients with blood VL blips have increased sCD163 levels?

3- Can the authors look at associated cell viral DNA in blood and run correlation with sCD163?

4- It would be valuable to have the absolute count of circulating monocytes expressing or not KI67 and look for any correlation with sCD163.

5- As sCD163 can be elevated in case of bacterial infection but infection/ inflammation can't be evaluated by looking at the leucocytes count in those HIV+ patients, it would be informative to have a blood marker to evaluate bacterial infection like LSP.

6- The choice of nomenclature to qualify the patients with macrophage activation is confusing and can be revisited. "at risk" suggest that the authors identified a consequence of this on going activation like "more at risk to rebound" or more likely to show early aging or anything related to on going activation. Since non of those symptoms was directly related to the group showing high level of sCD163, I would suggest finding another nomenclature to define the patient with macrophages activation.

7- Can similar profile (increase of sCD163) be found in patients who are not modifying their treatment?

Figure suggestion:

We should be able to visualize "at risk" and "no risk" population in figure #1.

Author Response

Dear reviewer,

Thank you for your comments, which certainly contributed to improve its quality.

On behalf of all co-authors, please find here the responses to your suggestions and the revised manuscript.

Kind regards

Matteo Vassallo

In this paper the authors are following inflammation and activation blood markers in patients who underwent cART modification from traditional Tri-therapy to dual therapy. The authors are reporting soluable inflammation markers quantification before cART switch, then at 6 month later and 18 to 24 months later. I would like to offer a few suggestions and have a few questions to hopefully improve this manuscript.

  • We started with 20 patients, but 6 were excluded because of non following up to 21 months. There is no details on how many time point the authors were able to get form those 6 patients but it would have been great to add them to the study if there is any data at the 6 months time point.

Thank you for the remark. This study was initially scheduled for a 6 months period of follow-up. All the 20 individuals completed such follow-up and results are those we previously published and mentioned in the Introduction session (reference 10). As a consequence of our results, we decided an extension of the study for continuing the follow-up until 24 months. Just 14 subjects completed such follow-up, for the other 6 we do have data about immune activation markers until 6 months, but with editor’s and reviewer’s permission, we think it is not interesting here to add the information we already published in another paper. If you think important we can add it.

  • Did the patients with blood VL blips have increased sCD163 levels?

Just one patient had two viral blips, defined by a low-level viremia, therefore it is difficult to find any correlation with sCd163 levels. Cases of very-low level viremia increased from 2 at baseline to 6 at the end of follow-up. Increases of sCD163 were higher in subjects with at least one of the following conditions: low CD4 nadir, previous AIDS or VLLV during the follow-up. However, VLLV alone did not correlate with the trajectory of sCD163. We added a sentence in the results session

  • Can the authors look at associated cell viral DNA in blood and run correlation with sCD163?

Unfortunately, we do not have any data about HIV-DNA and its correlation with sCD163. We added a sentence among the limitations

  • It would be valuable to have the absolute count of circulating monocytes expressing or not KI67 and look for any correlation with sCD163.

Unfortunately, we did not measure it. We added a sentence among the limitations

  • As sCD163 can be elevated in case of bacterial infection but infection/ inflammation can't be evaluated by looking at the leucocytes count in those HIV+ patients, it would be informative to have a blood marker to evaluate bacterial infection like LSP.

We included in the analysis the lipopolysaccharide binding protein (LBP) and we did not find any difference between subjects at high or low risk of immune activation. We then concluded that such macrophage activation is not associated with microbial translocation. We added a sentence in the discussion session

  • The choice of nomenclature to qualify the patients with macrophage activation is confusing and can be revisited. "at risk" suggest that the authors identified a consequence of this on going activation like "more at risk to rebound" or more likely to show early aging or anything related to on going activation. Since non of those symptoms was directly related to the group showing high level of sCD163, I would suggest finding another nomenclature to define the patient with macrophages activation.

We acknowledge that the term “at risk” could be confounding. We propose to replace it by the term “greater” or “lower” likelihood of immune activation

  • Can similar profile (increase of sCD163) be found in patients who are not modifying their treatment?

Thank you for the important question. The major limit of the study was the lack of a comparative arm, evaluating the trajectory of sCD163 in subjects who did not change therapy. With the lack of a comparative, someone could say that the increase in sCD163 is simply be associated with the physiological aging. However, some previous papers that now we added among references either did not find any change of sCD163 in treated patients or, in the case of Knudsen et al, found that each quartile increase of sCD163 over a period of 10 years was associated a 35% increased risk of death. We added some sentences in the discussion session about it.  

Figure suggestion:

We should be able to visualize "at risk" and "no risk" population in figure #1.

We modified Figure 1, adding the 2 trajectory of sCd163 in the two subgroups of patients

Reviewer 2 Report

Vassallo et al. report here the results of a clinical study named ‘Immunadapt’ which was a single-arm prospective study initiated in April 2019 with six- and 18-21 months follow-up time points. The study assessed the impact on immune activation markers of switching from a triple-drug to a dual-drug anti-retroviral treatment regimen in individuals with stable viral suppression. Other studies have demonstrated that two-drug regimen are not necessarily inferior to three-drug combination while reducing the complexity of therapy adherence and potential side effects for the patients. The authors assessed six established, relevant parameters related to immune activation and find an increase in sCD163 at both time points, suggesting increased activation of macrophages. The authors find that this is opposite to what happens with sCD14 which decreases although the changes didn’t reach significance in the present study. However, the authors should mention and discuss that MCP-1/CCL2 shows a very similar change and which clearly is a trend with a p value of 0.097 at 21 months. While the consequences of the increased macrophage activation are not clear yet, they raise interesting questions in terms of other health conditions that have been linked to macrophage activation, such as atherosclerosis. Overall the study is clearly designed and reported except for table 2. Headings for ‘Yes’ and ‘No’ in table 2 are unclear, specifically which columns belong to what heading. Although limited in size and the single arm design, the study adds interesting data needed for long-term management of HIV infection.

Author Response

Dear reviewer,

Thank you for your comments, which certainly contributed to improve its quality.

On behalf of all co-authors, please find here the responses to your suggestions and the revised manuscript.

Kind regards

Matteo Vassallo

Vassallo et al. report here the results of a clinical study named ‘Immunadapt’ which was a single-arm prospective study initiated in April 2019 with six- and 18-21 months follow-up time points. The study assessed the impact on immune activation markers of switching from a triple-drug to a dual-drug anti-retroviral treatment regimen in individuals with stable viral suppression. Other studies have demonstrated that two-drug regimen are not necessarily inferior to three-drug combination while reducing the complexity of therapy adherence and potential side effects for the patients. The authors assessed six established, relevant parameters related to immune activation and find an increase in sCD163 at both time points, suggesting increased activation of macrophages. The authors find that this is opposite to what happens with sCD14 which decreases although the changes didn’t reach significance in the present study. However, the authors should mention and discuss that MCP-1/CCL2 shows a very similar change and which clearly is a trend with a p value of 0.097 at 21 months. While the consequences of the increased macrophage activation are not clear yet, they raise interesting questions in terms of other health conditions that have been linked to macrophage activation, such as atherosclerosis. Overall the study is clearly designed and reported except for table 2. Headings for ‘Yes’ and ‘No’ in table 2 are unclear, specifically which columns belong to what heading. Although limited in size and the single arm design, the study adds interesting data needed for long-term management of HIV infection.

Thank you for your comments. We added some sentences in the results and discussion sessions in order to comment the similar trajectory of MCP-1/CCL2 and sCD14, in contrast with that of sCD163. We also modified the position of headings in Table 2